# Cyanobacteria use micro-optics to sense light direction

**Nils Schuergers[1†], Tchern Lenn[2], Ronald Kampmann[3], Markus V Meissner[3], Tiago Esteves[4,5,6], Maja Temerinac-Ott[7], Jan G Korvink[3], Alan R Lowe[8,9], Conrad W Mullineaux[2,7]\*, Annegret Wilde[1,10]**

[1]Institute of Biology III, University of Freiburg, Freiburg, Germany; [2]School of Biological and Chemical Sciences, Queen Mary University of London, London, United Kingdom; [3]Institute of Microstructure Technology, Karlsruhe Institute of Technology, Karlsruhe, Germany; [4]Instituto de Investigação e Inovação em Saúde, Universidade do Porto, Porto, Portugal; [5]INEB- Instituto de Engenharia Biomédica, Universidade do Porto, Porto, Portugal; [6]Faculdade de Engenharia da Universidade do Porto, Portugal; [7]Freiburg Institute for Advanced Studies, University of Freiburg, Freiburg, Germany; [8]London Centre for Nanotechnology, London, United Kingdom; [9]Institute for Structural and Molecular Biology, University College London and Birkbeck College London, London, United Kingdom; [10]BIOSS Centre for Biological Signaling Studies, University of Freiburg, Freiburg, Germany

**Abstract** Bacterial phototaxis was first recognized over a century ago, but the method by which such small cells can sense the direction of illumination has remained puzzling. The unicellular cyanobacterium *Synechocystis* sp. PCC 6803 moves with Type IV pili and measures light intensity and color with a range of photoreceptors. Here, we show that individual *Synechocystis* cells do not respond to a spatiotemporal gradient in light intensity, but rather they directly and accurately sense the position of a light source. We show that directional light sensing is possible because *Synechocystis* cells act as spherical microlenses, allowing the cell to see a light source and move towards it. A high-resolution image of the light source is focused on the edge of the cell opposite to the source, triggering movement away from the focused spot. Spherical cyanobacteria are probably the world's smallest and oldest example of a camera eye.

**\*For correspondence:** c.mullineaux@qmul.ac.uk

**Present address:** [†]Laboratory of Nanobiotechnology, Ecole Polytechnique Fédérale de Lausanne, Lausanne, Switzerland

**Competing interests:** The authors declare that no competing interests exist.

## Introduction

Many prokaryotes move directionally in response to a chemical or physical stimulus. However, it is generally assumed that bacteria are too small for direct sensing of a concentration gradient across the cell: instead they probe changes in stimulus concentration over time, as in the classic paradigm of flagella-mediated chemotaxis in *Escherichia coli* (reviewed by *Wadhams and Armitage, 2004*). When moving through a spatial concentration gradient of an attractant, *E. coli* cells experience temporal concentration changes, which they sense by employing a biochemical memory that directs a "biased random walk". Swimming along a straight path (run) alternates with random changes of direction (tumble). Tumbles become less frequent when cells sense a temporal increase in attractant concentration, introducing a bias to movement up a concentration gradient (*Berg and Brown, 1972*).

For phototrophic prokaryotes, light is the main source of energy but also potentially harmful, depending on intensity and wavelength. Unsurprisingly, many phototrophs can alter their movement in response to the light environment (reviewed in *Häder, 1987*). Bacterial phototaxis was first noted

**eLife digest** Cyanobacteria are blue-green bacteria that are abundant in the environment. Cyanobacteria in the oceans are among the world's most important oxygen producers and carbon dioxide consumers. *Synechocystis* is a spherical single-celled cyanobacterium that measures about three thousandths of a millimetre across. Because *Synechocystis* needs sunlight to produce energy, it is important for it to find places where the light is neither too weak nor too strong. Unlike some bacteria, *Synechocystis* can't swim, but it can crawl across surfaces. It uses this ability to move to places where the light conditions are better.

It was already known that *Synechocystis* cells move towards a light source that is shone at them from one side, which implies that the cyanobacteria can "see" where the light is. But how can such a tiny cell accurately detect where light is coming from?

Schuergers et al. tracked how *Synechocystis* moved in response to different light conditions, and found that the secret of "vision" in these cyanobacteria is that the cells act as tiny spherical lenses. When a light is shone at the cell, an image of the light source is focused at the opposite edge of the cell. Light-detecting molecules called photoreceptors respond to the focused image of the light source, and this provides the information needed to steer the cell towards the light. Although the details are different, and although a *Synechocystis* cell is in terms of volume about 500 billion times smaller than a human eyeball, vision in *Synechocystis* actually works by principles similar to vision in humans.

Schuergers et al.'s findings open plenty of further questions, as other types of bacteria may also act as tiny lenses. More also remains to be learnt about how the cyanobacteria process visual information.

in 1883 (*Engelmann, 1883*) and has been characterized in free-swimming phototrophs including purple bacteria and *Halobacterium* spp. (*Hildebrand and Dencher, 1975*; *Alam and Oesterhelt, 1984*). Cyanobacteria, which are oxygenic phototrophs, do not swim with flagella. Instead, various species exhibit "twitching" or "gliding" motility over moist surfaces (*Pringsheim, 1968*). This movement can be directed towards a light source, thus constituting true phototaxis (*Choi et al., 1999*; *Bhaya, 2004*; *Yoshihara and Ikeuchi, 2004*).

The model unicellular cyanobacterium *Synechocystis* sp. PCC 6803 (hereafter *Synechocystis*) has spherical cells about 3 μm in diameter and moves using Type IV pili (T4P) (*Bhaya et al., 2000*; *Yoshihara et al., 2001*). The location of the T4P extension motor PilB1 implies that pili are extended at the leading edge of the cell, and therefore that movement is generated by pilus retraction (*Schuergers et al., 2015*; *Wilde and Mullineaux, 2015*), as has been shown in other bacteria (*Merz et al., 2000*). It has recently been established that the motility of a filamentous cyanobacterium is also T4P-dependent, suggesting that this form of motility is widespread in cyanobacteria (*Khayatan et al., 2015*). One likely exception is marine *Synechococcus spp.*, which swims and exhibits chemotaxis without obvious surface appendages (*Willey and Waterbury, 1989*; *Ehlers and Oster, 2012*) apart from short spicules found in one of the motile *Synechococcus* strains (*Samuel et al., 2001*).

*Synechocystis* T4P-dependent phototaxis can be observed microscopically at the single cell level and macroscopically through the migration of cell colonies. Genetic studies have identified a number of photoreceptors that influence phototactic behavior under different light regimes (*Bhaya, 2004*). While *Synechocystis* harbors signal transduction systems for pilus biogenesis that are homologous to the chemotaxis system in *E. coli*, it lacks the CheR methyltransferase and the CheB methylesterase that are required in most chemotactic bacteria to sense temporal changes in attractant concentration (*Wuichet and Zhulin, 2010*). This suggests a different mode of directional control.

Previous studies of *Synechocystis* single cell phototaxis (*Choi et al., 1999*; *Chau et al., 2015*) have not addressed the question of how an individual cell might be able to perceive the direction of illumination. Here, we establish that individual *Synechocystis* cells can directly and accurately perceive the position of a unidirectional light source, and control their motility so as to move towards it. We then show that *Synechocystis* cells act as microlenses, and that the light intensity gradient across

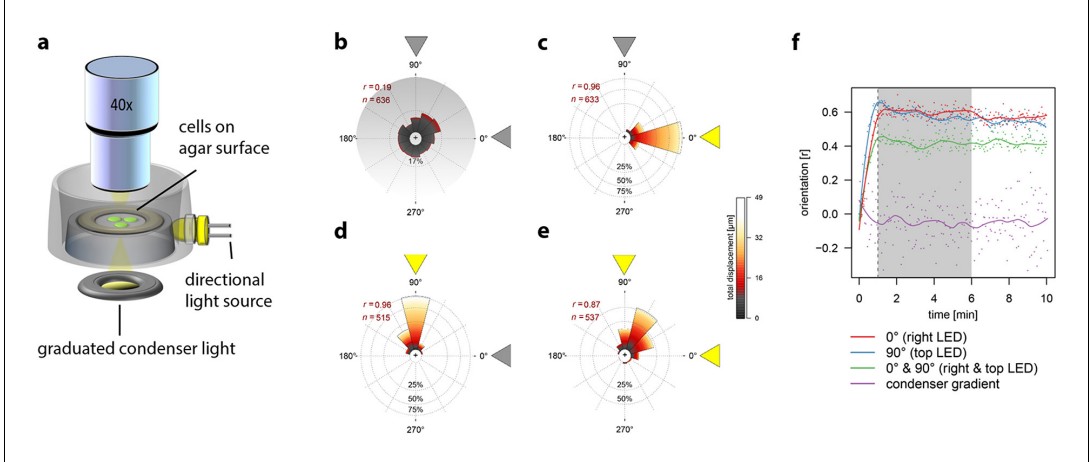

**Figure 1.** Movement of individual *Synechocystis* cells under different light regimes. Displacement over a 5 min time-frame was measured 1 min after the onset of illumination. The mean resultant length from a Rayleigh test (*r*) and the number of tracked cells (*n*) are shown. See also *Video 1*. All data were obtained with the same cells from a single continuous experiment. (**a**) Schematic diagram to illustrate the optical set-up for a light gradient projected onto the agar surface (as in b) and oblique directional illumination (as in c–e). (**b**) Cells moving in a white light gradient from 0–20 μmol photons m$^{-2}$ s$^{-1}$ over a 165 μm interval from the top to the bottom of the plot, showing no significant directional bias. (**c**) Illumination from an oblique RGB LED light source from the right, at intensity 10 μmol photons m$^{-2}$ s$^{-1}$. (**d**) Illumination from a similar light source placed orthogonally to that in (**b**). (**e**) Simultaneous illumination from both oblique light sources. (**f**) Correlation of cell movement with light direction, as a function of time after applying the light. The shading indicates the time window plotted in b–e. The *y*-axis shows the mean resultant length from a Rayleigh test (*r*) where 0 indicates random displacements and 1 indicates maximal clustering in the direction of illumination. LED, light emitting diode.

the cell due to this lensing effect is far greater than the effects of shading due to light absorption or reflection. Finally, we use highly-localized laser excitation to show that specific excitation of one side of the cell triggers movement away from the light, indicating that positive phototaxis results from movement away from an image of the light source focused on the opposite side of the cell. Essentially, the cell acts as a microscopic eyeball.

## Results

### *Synechocystis* phototaxis is based on direct light perception, rather than a biased random walk

Individual *Synechocystis* cells moving in two dimensions on an agarose surface in response to different light regimes were tracked microscopically to determine whether single cells are capable of direct perception of the position of a light source. First, we tested the response of cells to a light intensity gradient projected onto the surface from the microscope condenser (*Figure 1a*), using a gradient of white light from 0–20 μmol photons m$^{-2}$ s$^{-1}$, an intensity range that is relevant for positive phototaxis (*Choi et al., 1999*; *Bhaya, 2004*; *Chau et al., 2015*). However, the cells moved randomly without any significant directional bias (*Figure 1b*; *Video 1*). If *Synechocystis* phototaxis were based on a biochemical memory like *E. coli* chemotaxis, cells would perceive temporal changes in light intensity as they move across the surface through the light gradient and would then accumulate in regions of optimal light intensity. This did not occur (*Figure 1b*, *Video 1*). By contrast, when cells were illuminated by a unidirectional light source (RGB illumination at 10 μmol photons m$^{-2}$ s$^{-1}$) at an angle oblique to the surface (*Figure 1a*), the majority of motile cells switched direction within about 1 min, and then moved directly towards the light source (*Figure 1c,d,f*; *Video 1*). Under illumination from two equal-intensity orthogonal light sources, the majority of cells moved towards a point midway between the two light sources (*Figure 1e*; *Video 1*). These behaviors are not consistent with a run-and-tumble mechanism or any kind of biased random walk. In accord with previous studies (*Choi et al., 1999*; *Chau et al., 2015*), we conclude that individual cells can directly and accurately perceive the position of a light source and control their motility accordingly.

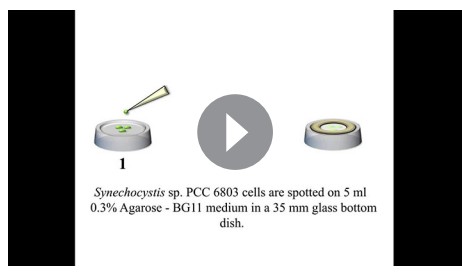

**Video 1.** Motility of *Synechocystis* cells under different illumination regimes. The video gives a schematic overview of the experimental set-up, followed by movement of cells in a projected light gradient, and with oblique illumination from two orthogonal directions, and then from both directions simultaneously. In each case, the raw video data is followed by the same movie clip with the tracks of cells superimposed. Time in minutes is indicated.

## Shading is minimal in single *Synechocystis* cells

*Synechocystis* cells contain a dense lamellar system of thylakoid membranes packed with photosynthetic complexes, and we initially postulated that light direction sensing depends on shading by the thylakoids, with the motility apparatus activated by brighter light at the illuminated side of the cell and deactivated at the shaded side of the cell. To assess the plausibility of this idea, we estimated the transmission spectrum of a single motile cell, scaling the absorption spectrum for a cell suspension according to the mean cellular pigment content. To ensure that pigment content was appropriate for phototactic cells, we used cells taken from moving colonies on motility plates (*Figure 2*). The transmission spectrum shows that even at peak absorption wavelengths, a single cell can absorb only about 20% of the photons that pass through it (*Figure 2*).

Although local pigment concentrations within the cell are quite high (*Figure 2*), the very short optical path length means that light absorption by the cell is low. Note that our estimate assumes a homogeneous distribution of pigments within the thylakoid region of the cell: in reality, pigment clustering will tend to decrease the cell absorbance due to enhanced self-shading of pigments (*Duysens, 1956*). Thus, 20% represents a maximum estimate of the proportion of photons that can be absorbed, and the light intensity gradient across the cell due to shading must be almost negligible. The direct measurement of single cell absorption spectra is technically challenging, and we are aware of only one such measurement in the literature for a cyanobacterium. *Sugiura and Itoh (2012)* show that the peak absorbance for a single cell of *Nostoc* sp. is about 0.04. This corresponds to a peak absorption of about 10% of the photons that pass through the cell, which is even lower than our estimate for *Synechocystis*.

## *Synechocystis* cells act as microscopic spherical lenses

Most light microscopy uses illumination orthogonal to the surface on which the sample rests, with the exception of dark-field microscopy, which uses oblique illumination from all sides. While investigating *Synechocystis* phototaxis, we observed cells instead with oblique illumination from one side only. Observations with this unusual illumination suggest a solution to the problem of directional light perception in *Synechocystis*. These images reveal that each cell acts as a microscopic spherical lens, focusing an intense light spot close to the *opposite* side of the cell from the light source and the direction of movement (*Figure 3a*). Images from two orthogonal light sources (as employed in one of the motility assays in *Figure 1e* and *Video 1*) are focused at different points on the cell periphery (*Figure 3a*), indicating that the cell can focus an image of its surroundings at the plasma membrane.

The images in *Figure 3a* do not give a quantitative picture of the lensing effect since the focused light spot is observed very indirectly via light reflected from the agar surface. Therefore, to quantify the lensing effect, we first employed a *Synechocystis* mutant that accumulates green fluorescent protein (GFP) evenly distributed in the periplasm (*Spence et al., 2003*). Thus, GFP can be used as local reporter of light intensity at the cell periphery (*Figure 3b*) since GFP fluorescence will be proportional to the local excitation light intensity. GFP fluorescence images with oblique laser excitation show a spot of light sharply focused at the opposite edge of the cell. Profiles of GFP fluorescence around the cell perimeter (*Figure 3c*) indicate that the mean ratio of the light intensity at the center of this focused spot to the intensity at the front of the cell facing the light source was 4.1 ± 1.5 (mean and standard deviation, $n$ = 13 cells from one representative experiment) and the observed mean full width at half maximum (FWHM) of the focused spot was 609 ± 30 nm (mean and standard deviation, $n$ = 11 cells). Since the measured FWHM for the point-spread function of the

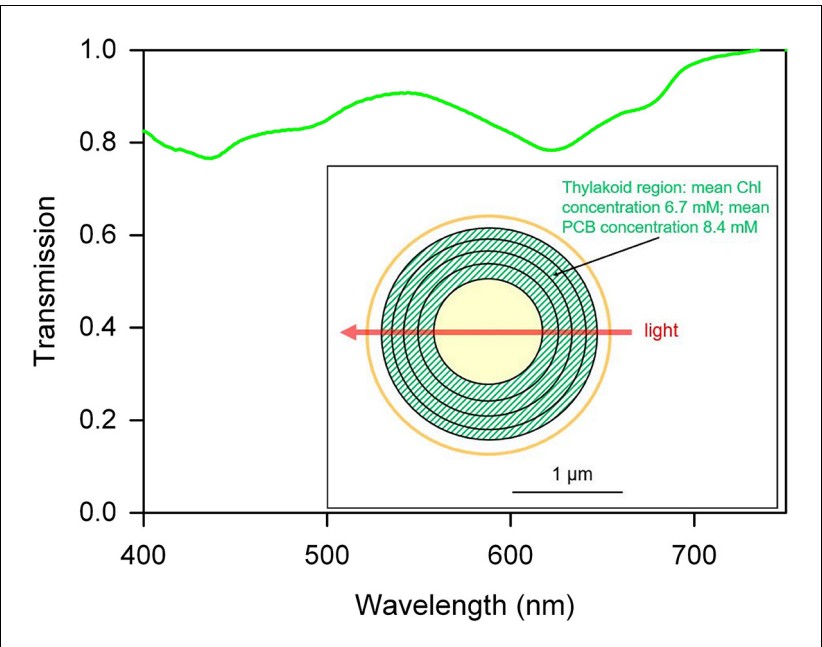

**Figure 2.** Estimated transmission spectrum of a single motile *Synechocystis* cell. Estimate obtained by scaling and converting the absorption spectrum for a suspension of cells from a moving *Synechocystis* colony, as detailed in Materials and methods. Pigments (including chlorophyll: Chl and phycocyanin-coupled phycocyanobilin: PCB) are assumed to be evenly distributed within the thylakoid region as shown in the diagram, and the estimated transmission is for a narrow beam of light passing straight through the center of the cell, with an optical path length through the thylakoid region of 1 μm. The spectrum represents a minimum estimate for transmission through the cell, since the estimate assumes a homogeneous distribution of pigments within the thylakoid region. In reality, inhomogeneous distribution of membranes and pigments will tend to decrease absorption due to enhanced self-shading (*Duysens, 1956*).

microscope was 270 nm, the true FWHM for the focused spot of 488 nm light at the cell periphery can be estimated to be about 550 nm. This experiment shows that for 488 nm light, lensing effects leading to concentration of light at the opposite side of the cell from the light source are overwhelmingly greater than shading effects, which would lead to higher light intensity on the illuminated side of the cell. Shading effects at other wavelengths in the range from 400 to 750 nm could be only marginally greater than at 488 nm (*Figure 2*).

The spatial resolution of the fluorescence measurement in *Figure 3b* is limited by the optical point-spread function of the microscope. Therefore, for higher-resolution measurement of near-field light perturbation by *Synechocystis* cells, we used a photolithographic method that gives a very high resolution image of the light pattern adjacent to the cell, although it does not provide such a direct and quantitative measurement of relative light intensity as the fluorescence imaging in *Figure 3b*. *Synechocystis* cells were adsorbed onto the surface of a photopolymer and collimated UV light (365 nm) was projected vertically down onto them (*Figure 3d*). The UV light induces cross-linking of the photopolymer. After development, the surface relief of the photopolymer gives a high-resolution replica of the light field around and beneath each cell, which we examined by atomic force microscopy (*Figure 3E*). *Synechocystis* cells produced distinctive near-field optical scattering patterns on the polymer surface with a remarkably sharp and intense peak beneath the center of each cell. We measured an FWHM of 281 ± 33 nm (mean and standard deviation, *n* = 6 from profiles at different angles across the representative image shown) (*Figure 3e*). This experiment indicates that near-UV light is focused to a spot with diameter less than the wavelength. *Figure 3e* also confirms that near-UV light is concentrated at the side of the cell opposite to the light source in a similar manner to light in the visible range. As with visible light (*Figure 2b,c*), the lensing effect predominates over any shading effects with near-UV light (*Figure 3e*).

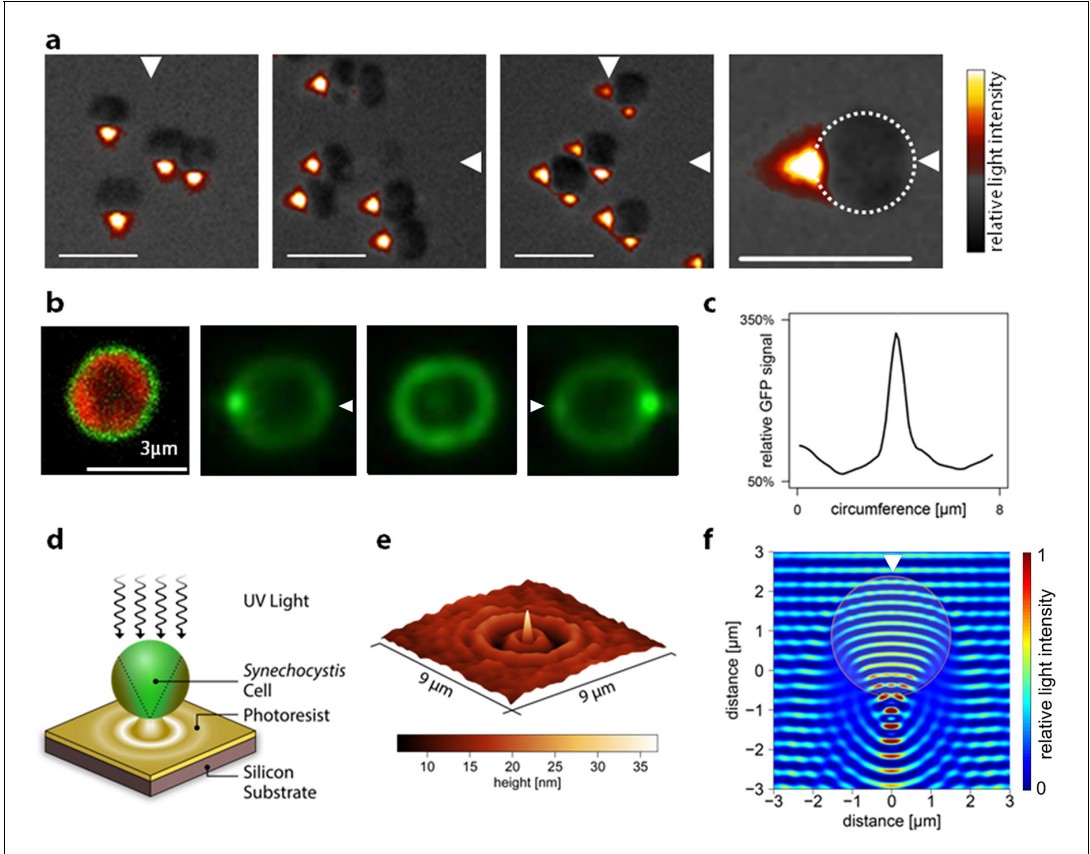

**Figure 3.** Micro-optic effects in cyanobacteria. (**a**) *Synechocystis* cells viewed with oblique illumination from the different directions shown, with enlarged image on the right. Scale-bars: 5 μm. (**b**) Images of periplasmic GFP fluorescence in *Synechocystis torA-gfp* (***Spence et al., 2003***). Left: two-channel confocal micrograph (GFP fluorescence in green; chlorophyll fluorescence in red) with GFP epifluorescence images of a single cell illuminated from the right, above and left (arrows indicate illumination direction; see ***Figure 3—figure supplement 1*** for the optical set-up). (**c**) GFP fluorescence profile around the cell circumference, extracted from the epifluorescence image in (**b**) with illumination from the right. The profile was taken in an anti-clockwise direction starting at the point nearest the light source. (**d**) Schematic illustration of the measurement by photolithography of near-field optical effects of a *Synechocystis* cell. (**e**) Height profile reconstructed from an AFM image of a photolithograph from the experiment illustrated in (**d**). (**f**) Finite difference time domain model of the light path (wavelength 365 nm) through a *Synechocystis* cell (illumination direction indicated by the arrow). The color scale indicates relative light intensity obtained by time-averaging the amplitude of the Poynting vector for the electromagnetic field. The wave patterns represent a snapshot of the oscillating electromagnetic field propagating through the model cell. GFP, green fluorescent protein.

The following figure supplement is available for figure 3:

**Figure supplement 1.** Optical set-up for oblique-angle excitation fluorescence microscopy to visualize periplasmic GFP fluorescence (***Figure 3c***).

To probe the physical basis for the lensing effect, we modeled light perturbation by the cell with finite difference time domain (FDTD) simulations, a method that uses the electromagnetic Maxwell equations without any geometrical simplifications (***Yee, 1966***). This electromagnetic description traces all observed effects back to the interference of incoming waves with waves scattered by their encounter with the object, which in this case was a simplified model of the cell as a microsphere with uniform refractive index. The method predicts a combination of effects that, on larger scales, are described as interference, refraction and internal reflection. We found that the observed near-field light pattern (***Figure 3e***) could be accurately reproduced by an FDTD simulation approximating the cell as a dielectric sphere with a diameter of 3 μm and a refractive index of 1.4 (***Figure 3f***). Micro-spheres with similar dimensions to *Synechocystis* have been experimentally shown to produce similar sharply focused light beams at the edge of the object opposite to the light source: these are termed "photonic nanojets" (***Ferrand et al., 2008***; ***Heifetz et al., 2009***).

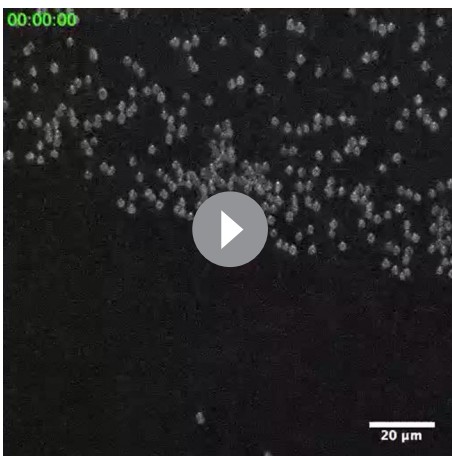

**Video 2.** Effects of a highly-focused laser spot on directional motility in *Synechocystis*. Cells are imaged by fluorescence from the photosynthetic pigments, and are moving towards an oblique LED light at the bottom of the frame: note the focused light spot at the rear edge of each cell. The superimposed red spot indicates the position of the laser, and time in min is shown at the top left. LED, light emitting diode.

### *Synechocystis* positive phototaxis is a photophobic response to excitation of one side of the cell

The micro-optic effects shown in *Figure 3* produce intensity differences across *Synechocystis* cells that are opposite in orientation and at least 20 times greater than those predicted from shading due to light absorption by the photosynthetic pigments (compare *Figure 3c* with *Figure 2*). *Figure 3a,b,c,e* provide direct experimental confirmation that light intensity is highest at the edge of the cell furthest from the light source. This suggests that the basis for directional light perception by *Synechocystis* should depend on the lensing properties of the cells, with positive phototaxis based on the cell moving away from the light spot focused at its periphery. We tested this idea by using a highly focused laser light spot (*Lowe et al., 2015*) as an alternative way to selectively illuminate one edge of the cell. *Synechocystis* cells moving directionally on an agarose surface towards a red (625 nm) light emitting diode (LED) light source were visualized by fluorescence from the photosynthetic pigments excited by the LED light. Focused spots of light at the rear periphery of the cell were again observed under this illumination regime (*Figure 4a*, *Video 2*), showing that the focused light spots extend into the thylakoid membrane region. The moving cells were allowed to encounter a spot of 640 nm laser light focused on the agarose surface (*Figure 4*; *Video 2*). The intensity gradient at the edge of the laser spot was steep enough to ensure specific exposure of one side of the cell to the light (*Figure 4a*). Whenever one edge of a cell encountered the edge of the laser spot, the cell changed direction to move away from the laser illumination (*Figure 4b*; *Video 2*). Cells did not cross the center of the intense laser spot, but instead changed direction when the laser light intensity at the front edge of the cell exceeded the intensity of the light spot focused by the cell at its rear periphery by a factor of about 2–10, as assessed from the brightness of fluorescence from the photosynthetic pigments (*Figure 4c*). In accord with our hypothesis, this shows that *Synechocystis* phototaxis is essentially a photophobic response to selective excitation of one side of the cell. The data in *Figure 4* and *Video 2* indicate that this photophobic response is increasingly strong with stronger localized excitation: thus, when the cells encounter laser light that is stronger than the focused light spot at the rear edge of the cell, they change direction to move away from the laser light.

## Discussion

### Directional light perception in *Synechocystis* depends on lensing, not shading

Here we have shown that *Synechocystis* cells act as very effective spherical microlenses that focus a sharp image of a light source at the opposite edge of the cell. This implies that positive phototaxis (i.e. movement towards a light source) is actually triggered as a negative response to the focused spot of light at rear periphery of the cell. We directly tested this idea by exposing cells to a spot of red laser light that was sharply focused enough to selectively excite one edge of the cell (*Figure 4*, *Video 2*). As predicted by our hypothesis, cells moved away from localized laser excitation that was only slightly brighter than the focused image of the light source (*Figure 4*, *Video 2*). This result is the opposite of what would be expected from a "shading" model for directional light perception, which would predict that light in this intensity range should attract the cells.

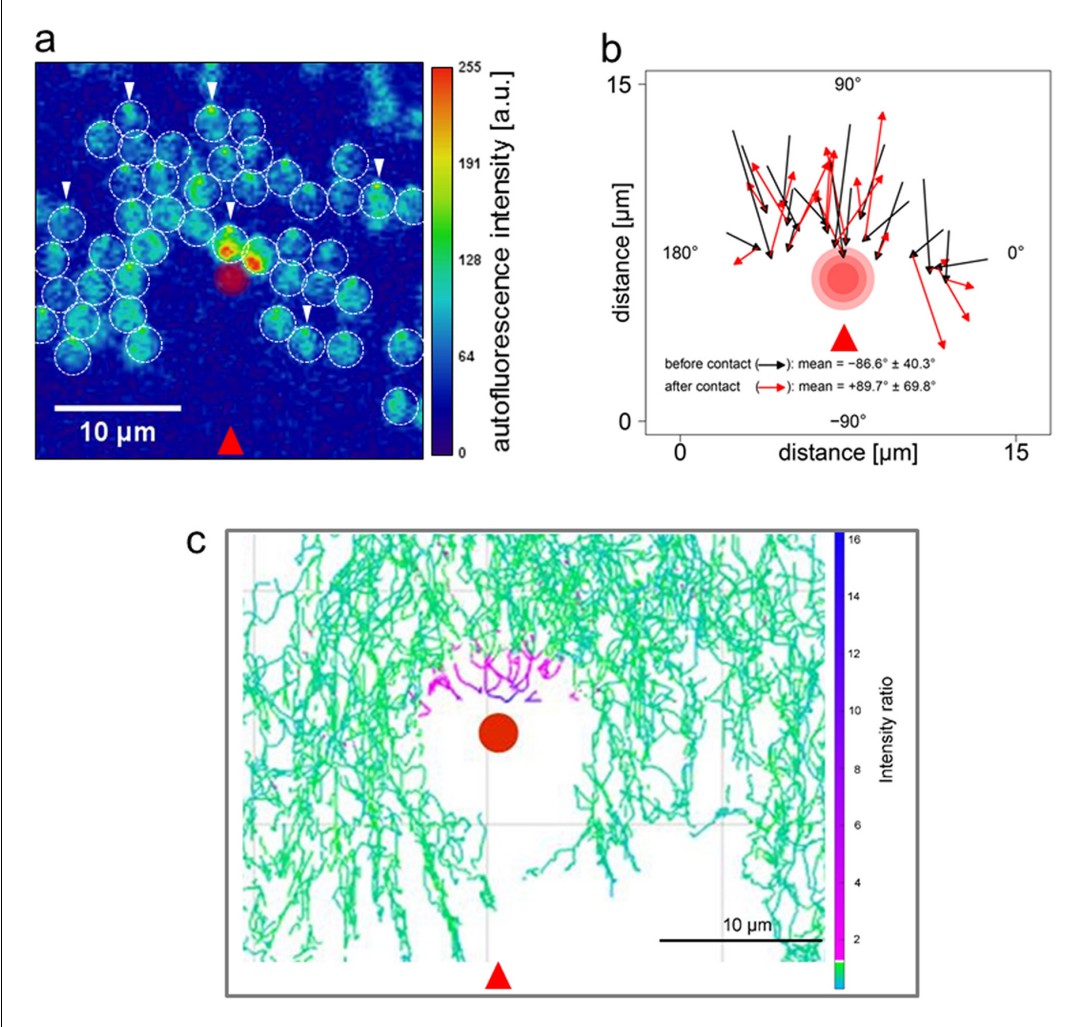

**Figure 4.** Direction switching triggered by specific excitation of one edge of the cell. *Synechocystis* cells were moving in response to an oblique 625 nm light source from the direction indicated by the red arrows and allowed to encounter a spot of 640 nm laser light focused on the agarose surface. See *Video 2* for the full data and *Figure 4—figure supplement 1* for the optical set-up and the intensity profile of the laser spot. (a) False color image of fluorescence from the photosynthetic pigments, with the laser spot indicated by the red circle. The broken white lines indicate the approximate cell boundaries and the white arrows highlight examples of the focused images of the light source at the rear edge of the cell. Cells approaching the laser spot show strong selective excitation of the leading edge of the cell. (b) Direction switching triggered by contact with the edge of the laser spot. The arrowed lines indicate net displacements of representative cells over time windows of 132 s before and after closest approach to the laser spot. The mean orientation of the tracks ( ± standard deviation, *n* = 29) is shown. (c) Light intensity required to reverse the path of *Synechocystis* cells. Tracks of the mid-points of individual cells are shown over a 30 min time window, with a color scale indicating the ratio of laser intensity to the intensity of the phototactic LED light focused on the cell. Purple color indicates tracks of cells in which autofluorescence induced by the laser exceeds autofluorescence induced by the light focused on the cell for phototaxis by at least two-fold. LED, light emitting diode

The following figure supplement is available for figure 4:

**Figure supplement 1.** Optical set-up used to measure the response of *Synechocystis* cells to a highly focused laser light spot (*Figure 4*; *Video 2*).

## Directional control of motility in *Synechocystis*

*Synechocystis* motility depends on the extension, adhesion and retraction of T4P (*Bhaya, 2004*), which is powered by the motor ATPases PilB and PilT (*Merz et al., 2000*). We previously examined the localization of one of these motor ATPases in *Synechocystis* with GFP-tagging (*Schuergers et al., 2015*). We were unable to generate a strain with functional GFP-tagged PilT, but a strain expressing PilB1-GFP in a Δ*pilB1* null background was motile and capable of phototaxis, albeit with lower efficiency than the wild type (*Schuergers et al., 2015*). Imaging of PilB1-GFP in

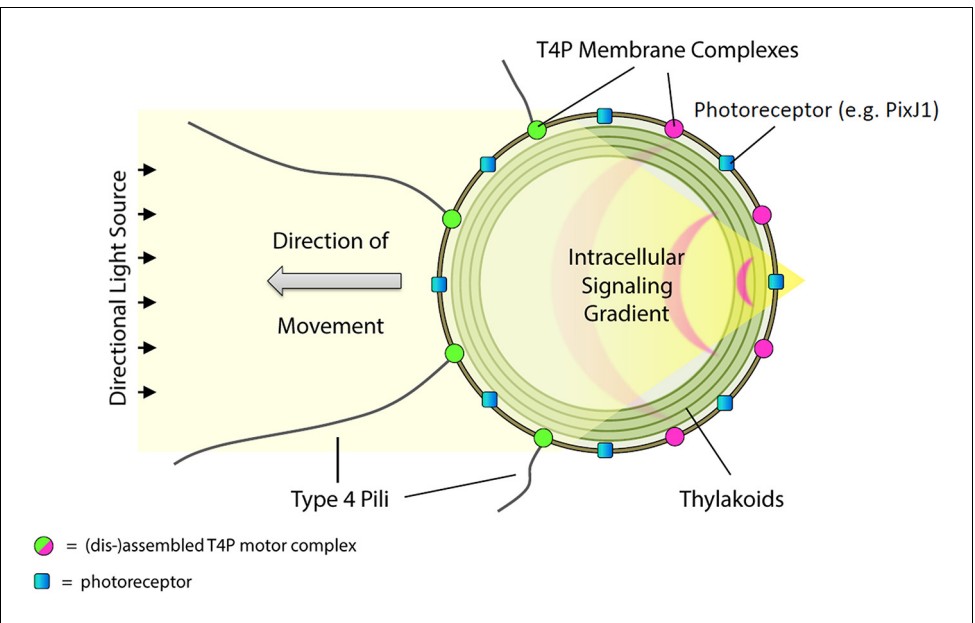

**Figure 5.** Model for control of positive phototaxis in *Synechocystis*. Directional illumination of the cell produces a sharply focused and intense spot of light (resembling a photonic nanojet) at the cell periphery on the opposite side from the light source. The focused spot is perceived by photoreceptors in the cytoplasmic membrane (for example PixJ1) triggering signal transduction via CheY-like response regulators that locally inactivates the T4P motility apparatus, dispersing T4P components including the extension motor PilB1. Consequently, patches of the motor proteins can only form on the side of the cell facing the light source. Pili are extended and retracted at this side of the cell, which therefore moves towards the light. T4P, Type IV pili.

*Synechocystis* shows that it is localized in a crescent-like zone of the plasma membrane at one side of the cell, and that these patches can dynamically relocate to other areas of the membrane (*Schuergers et al., 2015*). For cells without directional illumination, random relocation of the PilB1-GFP patch frequently occurs within a 5 min window. The direction of motility in *Synechocystis* strongly correlates with the position of the PilB1 patch, indicating that direction is determined by the localization of PilB1, probably in concert with other T4P components such as PilT (*Schuergers et al., 2015*). In *Myxococcus xanthus*, both PilB and PilT show dynamic relocalization between the two cell poles, on comparable timescales (*Bulyha et al., 2009*).

## Phototactic signal transduction in *Synechocystis*

It is clear that multiple photoreceptors are involved in *Synechocystis* phototaxis, with light sensors that absorb at different wavelengths and trigger either positive or negative phototaxis (*Ng et al., 2004*). In addition, motility is modulated by the second messengers cyclic diguanylate monophosphate (c-di-GMP) (*Savakis et al., 2012*) and cyclic adenosine monophosphate (cAMP) (*Terauchi and Ohmori, 1999*; *Bhaya et al., 2006*). However, the best-characterized candidate for a directional photoreceptor for positive phototaxis is PixJ1 (also known as TaxD1). Mutants in which the *pixJ1* gene is inactivated show only negative phototaxis (*Yoshihara et al., 2000*; *Bhaya et al., 2001*). PixJ1 is a cyanobacteriochrome with two transmembrane domains, a chromophore-binding domain and a domain with similarity to the methyl-accepting chemotaxis proteins of enterobacteria at the C-terminus (*Bhaya et al., 2001*; *Yoshihara and Ikeuchi, 2004*). Proteomic studies with cell fractionation indicate that PixJ1 is located in the plasma membrane (*Pisareva et al., 2007*) and this is an important consideration for our model.

Downstream signal transduction from PixJ1 likely involves the products of neighboring genes in the *tax1* locus, since mutants lacking these genes (apart from *pixI*) also show only negative phototaxis (*Bhaya et al., 2001*; *Yoshihara et al., 2000*). The respective gene products include homologs of the *E. coli* chemotaxis signal transducers CheW, CheA and CheY (*Bhaya et al., 2001*; *Yoshihara and Ikeuchi, 2004*). A likely signal transduction pathway would proceed through light

activation of PixJ1 (sll0041) that regulates autophosphorylation of the CheA homolog sll0043 (PixL) via the CheW homolog sll0040 (PixI). PixL could transfer the phosphate group to the CheY-like response regulators sll0038 (PixG) and/or sll0039 (PixH). By analogy with the *E. coli* chemotaxis system (reviewed by *Sourjik and Wingreen, 2012*), these response regulators might interact directly with the *Synechocystis* motility apparatus.

## A model for directional light perception and phototaxis

Combining our current results with the previous study on PilB1 localization and motility in *Synechocystis* (*Schuergers et al., 2015*) and the likely scheme for signal transduction discussed above leads to a simple model for control of positive phototaxis in *Synechocystis,* which is illustrated in *Figure 5*. The bright focused image of the light source is perceived by PixJ1 in the plasma membrane, resulting in local changes in the phosphorylation status of the response regulators PixG and/or PixH, which leads to local inactivation of the T4P motility apparatus and dispersal of the motor proteins (PilB1 and likely also PilT). The motility apparatus therefore assembles at the side of the cell facing the light, resulting in movement towards the source (*Figure 5*). The 1 min timescale for direction switching in *Synechocystis* (*Figure 1f*) is consistent with the rapid kinetics of relocalization of PilB1 patches (*Schuergers et al., 2015*). Responses to multiple light sources (as in *Figure 1e* and *Figure 4*) can be explained if it is assumed that the dispersal signal is graded with light intensity. Thus when a cell is exposed to a stronger light source at one edge (as with cells encountering the focused laser spot in *Figure 4* and *Video 2*), the motility apparatus is always most strongly inactivated in the region of strongest local illumination.

## Vision in *Synechocystis*

The model in *Figure 5* implies that, essentially the *Synechocystis* cell functions as a microscopic eyeball, with the spherical cell body as the lens and the cytoplasmic membrane as the retina.

From the observed dimensions of the spot of 488 nm light focused by *Synechocystis* cells (*Figure 3c*) we can estimate that *Synechocystis* "vision" has an angular resolution (FWHM) of about 21°, essentially limited by light wavelength and the area of the plasma membrane, which is tiny in comparison to an animal retina. However, this resolution is sufficient to incorporate quite complex spatial information into a 360° image of the cell's surroundings, and our data in *Figure 1e* indicate that the cell can integrate information from distinct and spatially separated light sources. The directional motile responses of the cells (*Figure 1c,d*) show a distribution of displacement angles with FWHM ~30°. This is less accurate than the initial imaging of the light source, which suggests an unsurprising degree of spreading and noise during the signal transduction that comes between initial light perception and the response of the motility apparatus.

## Wider implications

Our model (*Figure 5*) implies that the maximum efficiency of directional light sensing will be achieved in a spherical cell. Among the unicellular cyanobacteria, a number of species of the genera *Synechocystis* and *Gloeocapsa* are described as motile and phototactic, and these species do indeed have spherical or near-spherical cells (*Rippka et al., 1979*). However, the rod-shaped unicellular thermophilic cyanobacterium *Thermosynechococcus elongatus* is also motile and appears capable of direct perception of light direction (*Kondou et al., 2001*). The *Thermosynechococcus* homolog of *Synechocystis* PixJ1 (TePixJ) is clustered at the cell poles (*Kondou et al., 2002*). For our model to be applied to *Thermosynechococcus*, the cell would need a method to concentrate light at the pole furthest away from the light source. Micro-optic effects in rod-shaped cells such as those of *Thermosynechococcus* need further investigation.

Cyanobacteria are not the only unicellular organisms capable of directional light perception. *Rhodospirillum centenum* cells form phototactic swarm colonies, with an unknown mechanism of light direction sensing (*Ragatz et al., 1995*). Phototaxis in the larger cells of some unicellular eukaryotic green algae is mediated by clusters of photoreceptors known as eyespots, coupled with complex shading and reflecting devices (*Kreimer, 2009*). Lensing effects are implicated in phototactic orientation and phototropic responses in some other eukaryotes (*Shropshire, 1962*; *Häder and Burkart, 1983*). However, focused imaging at the cell periphery in the tiny spherical cells of *Synechocystis* represents a particularly elegant and compact solution to the problem of directional light-

perception, that is probably more ancient than the eukaryotic systems. Spherical cyanobacteria may have been the first organisms to see their world. The micro-optic effects that we observe in cyanobacterial cells are relevant not only to phototaxis but also to photosynthesis, since light distribution within the cell is so strongly affected. Future work will also need to explore the biological implications of optical lensing in non-photosynthetic bacteria, since such micro-optic effects are likely to be widespread in bacteria with cells of the appropriate size and shape.

## Materials and methods

### Strain, growth conditions and phototaxis assay

We used a motile sub-strain of the original *Synechocystis* sp. PCC 6803 wild type from the Pasteur Culture Collection, acquired from the lab of S. Shestakov (Moscow) in 1993. This sub-strain (PCC-M) was recently characterized by complete genome sequencing (*Trautmann et al., 2012*). *Synechocystis* cells were grown for at least 24 h on motility medium (0.3% (w/v) agarose in BG11 medium (*Stanier et al., 1971*) supplemented with 0.2% glucose and 10 mM Tris(hydroxymethyl)methyl-2-aminoethanesulfonic acid (TES) (pH 8.0) buffer) in dark boxes with a one sided opening leading to a directional illumination of 5–10 µmol photons m$^{-2}$ s$^{-1}$ (Phillips MASTER TL-D Super 80 18W/840, Philips GmbH Market DACH, Germany).

### Video measurement of single cell motility

Cells from the moving front of a colony on a motility plate were resuspended in fresh BG11 medium and 3 µl aliquots were directly spotted on top of 5 ml motility medium in 35 mm glass-bottom plates. When liquid droplets were no longer visible (5–10 min) a cover-slip was carefully placed on top of the cells and the surrounding surfaces of the plates were covered with a silicone ring to minimize surface oscillations and evaporation. Time-lapse videos were captured at room temperature (ca. 22°C) using an upright Nikon Eclipse Ni-U microscope (Nikon Instruments, Germany) fitted with a 40× objective (numerical aperture 0.75). Gradient illumination was with white light from the microscope condenser lamp. RGB LEDs (470/525/625 nm at equal intensity) (World Trading Net GmbH, Germany) for directional illumination were mounted in boreholes of a black plastic cylinder surrounding the motility plates. Light intensities were measured with a LiCOR light-meter with a planar quantum sensor (LI-COR Biosciences GmbH, Germany) with a detection window from 400–700 nm. Single cell movement was captured at 1 frame per 3 s. Unless otherwise stated, the exposure time was 200 ms to reduce the background light needed for visualizing the cells.

### Single cell tracking and data analysis

We developed the BacteriaMobilityQuant software (https://web.fe.up.pt/~dee11017/software/BacterialMobilityQuant.zip) implemented in MATLAB for tracking single cells in time lapse videos. It is based on a detection-association tracking approach that relies on the Laplacian of Gaussian filter (LoG) for the task of bacteria detection (*Esteves et al., 2013a*; *Esteves et al., 2013b*). The LoG filter is based on the image scale-space representation to enhance the blob-like structure as introduced by Lindeberg (*Lindeberg, 1994*). The scale of the LoG filter is set to the expected range of the bacterial radius. The bacteria detection is performed by finding local maxima of LoG response in the input image (number of patches for local maxima detection controlled manually by a parameter). The detected maxima enable the estimation of bacteria location. We performed bacteria tracking based on the spatial distance between detections in consecutive frames (*Esteves et al., 2013b*). Data analysis of the raw tracks was done using R software (R Core*Team, 2013*) implementing the CIRCULAR package (*Agostinelli and Lund, 2013*). To eliminate artifacts due to wrong mapping of dividing cells or cells in densely populated areas, only cells that could be tracked for at least 25 consecutive frames with an average velocity below 0.4 µm s$^{-1}$ and a maximum displacement of less than 8 µm between two frames (3 s) were considered. Cells with an average velocity lower than 0.05 µm s$^{-1}$ were regarded as immotile and discarded. Displacement and orientation were calculated for all cells that could be continuously tracked for 60 frames, during 5 min after the onset of the directional light. All the data shown in *Figure 1* are from a single continuous experiment to ensure the reliability of quantitative comparisons. The experiment is representative of >10 such experiments carried out under comparable conditions.

## Estimate of a single cell transmission spectrum

*Synechocystis* cells were scraped from a moving colony on a motility plate and resuspended in fresh BG11 medium. An absorption spectrum for the culture was recorded in an Aminco DW2000 spectrophotometer (Olis Inc, Bogart, GA), which is equipped with a wide detection window to minimize distortion due to light scattering. After subtraction of the optical density due to light scattering at 750 nm, the absorption spectrum was deconvoluted into chlorophyll and phycobilin components and chlorophyll and phycocyanobilin concentrations were estimated according to the formulae of *Myers et al. (1980)*. Cell density in the suspension was estimated by counting in a hemocytometer, leading to estimates of $2.94 \times 10^7$ chlorophyll molecules and $3.70 \times 10^7$ phycocyanin-coupled phycocyanobilin molecules per cell. These numbers are slightly higher than previously estimated (*Mann et al., 2000*); however, this study used different growth conditions. Effective pigment concentrations in the thylakoid membranes were then estimated by approximating the multiple thylakoid lamellae around the periphery of the cytoplasm as a hollow sphere with inner diameter 1 μm and outer diameter 2 μm, corresponding to the dimensions estimated from fluorescence micrographs. Pigments were assumed to be evenly distributed within the hollow sphere. A narrow light beam passing straight through the center of the cell would then have a total path-length of 1 μm through an effective chlorophyll concentration of 6.7 mM and an effective phycocyanobilin concentration of 8.4 mM. This would give a peak single cell absorbance of 0.106 at 620 nm from the published extinction coefficients (*Myers et al., 1980*). To estimate the full single cell absorption spectrum, the absorption spectrum for the culture was scaled appropriately. The scaled spectrum was converted into the transmission spectrum shown in *Figure 2*. Note that the calculation considers only light passing straight through the center of the cell and ignores any effects due to refraction and interference. In reality, the photosynthetic pigments will not be evenly distributed in the thylakoid region as we had to assume for the calculation, but rather clustered into membrane layers, reaction centers and light-harvesting complexes. Furthermore, the membranes themselves are usually not as regular and symmetrical as assumed in our model. A homogeneous distribution of pigments will maximize the absorption, since inhomogeneous clustering is liable to decrease absorption due to enhanced mutual shading of pigments (*Duysens, 1956*). Therefore, the spectrum shown represents a minimum estimate of light transmission through a single cell.

## Fluorescence measurements with oblique excitation

The *Synechocystis torA-gfp* mutant, expressing GFP fused to the TorA leader sequence for export to the periplasm, was previously described (*Spence et al., 2003*). Cells were grown in liquid culture to $OD_{750nm}$ ~1.0 in BG11 medium supplemented with 50 μg ml$^{-1}$ spectinomycin. 100 μl of culture was added to a chamber in an ibiTreat-coated 8-well μ-slide from ibidi and cells were left for 10 min to settle at the bottom. Suspended cells and excess medium were removed from the chamber, leaving adhered cells and a thin film of liquid, with some areas of the chamber bottom appearing dry. The samples were imaged immediately. Microscopy was performed using a modified version of the laser-spot time-lapse microscope (see *Figure 3—figure supplement 1*). Fluorescence excitation was achieved by near-TIRF ("highly inclined"/"oblique") illumination of the samples. Briefly, a fiber-coupled continuous wave (CW) diode laser (Toptica iChrome HP, 488 nm, Toptica, Germany) was injected into the epifluorescence port of the microscope via the dichroic mirror. The fiber, collimator, quarter-wave plate and TIR-lens (Thorlabs AC254-200-A-ML, Thorlabs, UK) were mounted on a translation stage and positioned such that the focused beam was centered at the back focal plane (BFP) of the Olympus 100× objective lens (UPON TIRF 1.49 NA oil) (Olympus, Japan). By adjusting the translation stage, the position of the beam could be translated across the BFP to adjust the inclination angle of the beam through the sample. Images were captured using the green channel using a GFP filter set (Semrock FF01-520/35-25, Semrock, Rochester, NY) and an exposure time of 100 ms. Average images were generated by frame averaging over 1–2 s. Measurements were taken from a single experimental run, representative of images recorded for three separate cultures of *Synechocystis torA-gfp*. Fluorescent profiles for spot intensity and FWHM measurements were obtained from the intensity profile of hand-fitted spline curves to the circumference of the cell image starting at the edge opposite to the focus spot (proximal to illumination source) and tracing in an anti-clockwise direction back to the origin. The line width was 3 pixels and the analysis was done in ImageJ.

## Photolithography

Permanent EPON epoxy-based photosensitive resin (SU-8 3000 series, MicroChem, Westborough, MA) with outstandingly low absorption in the near-UV range (*LaBianca and Gelorme, 1995*) was applied on pre-cleaned silicon wafers by spin coating. SU-8 3005 was coated at a final rotation speed of 4000 rpm for 30 s, after which the photosensitive film was subjected to a soft bake at 95°C for 5 hr. Droplets (4 μl) of an exponentially growing *Synechocystis* culture in BG11 medium were dispensed on top of the 5 μm thick photoresist. The BG11 medium evaporated under standard cleanroom conditions after 30 min, leaving the *Synechocystis* cells on top of the wafer. Flood exposure of the photopolymer was performed at a center wavelength of 365 nm (MA6 exposure system, Karl Suess, SÜSS MicroTec AG, Germany). The exposure dose of 85 mJ cm$^{-2}$ was found by a lithographic series, in order to achieve sharp rendering of scattering patterns on the SU-8 film. However, the SU-8 layer was not exposed entirely and rendered a height-dependent cross-linked structure similar to grayscale lithography (*Gal, 1994*). Height profiles of the scattering pattern were obtained by atomic force microscopy (tapping-mode, Dimension Icon, Bruker Nano Surfaces Division, Santa Barbara, CA). Completely exposed SU-8 patterns had a median roughness of $R_a$ = 3.2 nm. From the scattering patterns, a mean of six axially symmetric profiles was computed in order to average out the influence of surface roughness.

## Modeling of micro-optic effects

The optical field distribution was computed by the FDTD method. The algorithm solves the time-dependent Maxwell curl equations by using discrete time steps and leads to a time-resolved evolution of the spatial electromagnetic field distribution (*Yee, 1966*). We used a freeware FDTD implementation by Schmidt (*Schmidt, 2013*). The material interface geometry, including assignment of refractive index values to material regions, as well as the incoming light source properties were required to initiate the simulation process. The subsequent FDTD solution process was controlled by specifying a space and time grid ($\Delta x = \Delta y = 10$ nm, $\Delta t = 10^{-8}$ s).

## Motility assays with localized laser excitation

Wild-type motile *Synechocystis* cells were freshly plated in a line on motility plates and grown overnight at 30°C with directional illumination (~10 μmol photons m$^{-2}$ s$^{-1}$) in a standing incubator. Fingers of cells projecting 0.3–0.5 cm were seen after ~16 hrs. The leading edges of the fingers were collected and resuspended in 20–50 μl of BG11 medium and spotted onto freshly made motility plates and left to adsorb. Blocks of the motility agarose containing the spots were excised from the plate and inverted onto 35 mm glass-bottomed tissue-culture dishes for imaging in a custom-built inverted microscope. The time lapse microscope (see *Figure 4—figure supplement 1*) was constructed on the base of an Olympus IX81 microscope, using an Olympus 40× (UPLANFLUOR 0.75 NA Air) objective lens. Two air-cooled EMCCD cameras (Andor iXon Ultra DU-897U-CS0-#BV, Andor, UK) were coupled to the camera port of the microscope via a magnifying relay. The cameras were simultaneously triggered using a TTL pulse from an external digital-to-analog (D/A) converter (Data Translation DT9834, Data Translation, Germany), with an exposure time of 50 ms and a frequency of 0.25 Hz. A dichroic mirror (Semrock FF560-FDi01-25x36) in the Fourier plane of the camera relay split the emission into red (fluorescence) and green (transmission bright-field imaging) channels. Each channel had a separate band pass filter (Semrock BLP01-647R-25 and Semrock FF01-520/35-25). The focused laser spot was generated by expanding a collimated beam from a fiber-coupled CW diode laser (Toptica iChrome HP, 640 nm), injected via the epifluorescence port of the microscope and directed to the objective lens via a dichroic filter (Semrock Di01-R405/488/561/635-25x36). The beam was expanded to overfill the back aperture of the objective lens to achieve a diffraction-limited spot at the focal plane. A combination of neutral density filters and a half-wave plate and polarizing beam splitter were used to adjust the power of the laser to approximately 0.01 μW as measured using a power meter. A 1:1 lens pair (Thorlabs AC254-100-A-ML) with one lens mounted on a Z-translation stage was used to adjust the axial position of the focus at the sample plane. A quarter-wave plate was used to circularly polarize the beam before it was injected into the microscope. Bright-field trans-illumination was performed by fiber-coupling a 530 nm LED (Thorlabs M530F1) into a multimode fiber and imaging the magnified end of the fiber at the sample plane using a condenser lens. The LED was TTL triggered via the camera acquisition to reduce light

exposure to the cells, resulting in a synchronized 50 ms pulse during acquisition. A third LED (625 nm, 3 mW) was mounted close to the sample at an oblique angle to provide directional illumination for the motility assay. Light intensities were measured with a silicon photodiode-based power meter (Thorlabs S120C and PM100D). Acquisition was performed over 30 min. All the data shown in *Figure 4* are from a single continuous experiment, representative of 5 such experiments carried out under comparable conditions.

## Acknowledgements

We thank Senta Schauer (IMT, Karlsruhe Institute of Technology) for performing the AFM measurements, Brian Barnhart for help with the preparation of *Video 1*, and Giulia Mastroianni and Dennis J. Nürnberg for preliminary work that could not be included in the final paper.

## Additional information

### Funding

| Funder | Grant reference number | Author |
|---|---|---|
| Biotechnology and Biological Sciences Research Council | BB/J016985/1 | Tchern Lenn<br>Conrad W Mullineaux |
| European Research Council | 290586 | Ronald Kampmann<br>Markus V Meissner |
| Fundação para a Ciência e a Tecnologia | SFRH/BD/80508/2011 | Tiago Esteves |
| Bundesministerium für Bildung und Forschung | FKZ0316185 | Maja Temerinac-Ott |
| Medical Research Council | MR/K015826/1 | Alan R Lowe |
| Freiburg Institute for Advanced Studies, Albert-Ludwigs-Universität Freiburg | | Conrad W Mullineaux |

The funders had no role in study design, data collection and interpretation, or the decision to submit the work for publication.

### Author contributions

NS, ARL, CWM, Conception and design, Acquisition of data, Analysis and interpretation of data, Drafting or revising the article; TL, Conception and design, Acquisition of data, Analysis and interpretation of data; RK, AW, Conception and design, Analysis and interpretation of data, Drafting or revising the article; MVM, Acquisition of data, Analysis and interpretation of data, Drafting or revising the article; TE, MT-O, Conception and design, Analysis and interpretation of data; JGK, Conception and design, Drafting or revising the article

### Author ORCIDs

Jan G Korvink, http://orcid.org/0000-0003-4354-7295
Alan R Lowe, http://orcid.org/0000-0002-0558-3597

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
