## [Decision Letter]

Thank you for submitting your work entitled "Cyanobacteria use micro-optics to sense light direction" for consideration by *eLife*. Your article has been reviewed by three peer reviewers, one of whom is a member of our Board of Reviewing Editors. The evaluation has been overseen by the Reviewing Editor and Richard Losick as the Senior Editor. Two of the three reviewers have agreed to reveal their identity: Fred Rieke (Reviewing Editor and peer reviewer) and Carol Dieckman (peer reviewer).

The reviewers have discussed the reviews with one another and the Reviewing Editor has drafted this decision to help you prepare a revised submission.

This paper reaches an interesting and novel conclusion – that phototaxis in the cyanobacterium *Synechocystis* operates via the cell acting as a microlenses. All reviewers felt this conclusion was well supported by the data presented, particularly the laser spot experiments in Figure 4. Nonetheless, several issues need attention before we can reach a final decision on the paper. Those identified by several reviewers are summarized below:

1) The paper would benefit from a more detailed description of the signal transduction process between light absorption and the pili, as well as more details about how movement of the pili lead to cell movement. This should include a discussion of what is known about the photoreceptors themselves – e.g. what is their distribution? This could be added to the figure showing the proposed model.

2) Figure 1, Figure 2 and Figure 3 all lack some details that will be important for a broad readership. Suggestions for specific things to include are in the individual comments from the reviewers.

3) The *Thermosynechococcus* data in Figure 3 was distracting given the brevity with which it is presented. It needs either to get expanded or removed.

Individual reviewer comments:

*Reviewer #1:*

This paper presents a very interesting idea about the mechanisms underlying phototaxis in cyanobacteria. The central idea – that the cell membrane acts as a lens to focus light on the opposite side of the cell and that this leads to changes in directed movement – is well supported by several independent analysis approaches. There are some areas in which I think the paper can be improved. Specific comments follow.

Light gradient experiment. The explanation of the light gradient experiment (Paragraph three, Introduction) is confusing, and I did not fully understand it until watching the movie. I would suggest including a schematic like that in the movie in the main text to make sure the light configuration is clear. It might help to state that the bacteria (presumably) try to swim towards the condenser.

*Thermosynechococcus*: this is an interesting addition to the paper, but is quite underdeveloped. As a consequence it raises more questions than it answers – e.g. can they still integrate two light stimuli and if so how? Some quantification of where the spots are on the cell relative to the orientation of the light would strengthen this section. Alternatively, I think it could be left out without causing the main message to suffer.

Dispersal mechanisms (Paragraph four, Discussion): how is Figure 4 integration? Similarly I was not sure why you assume that the dispersal signal occurs at the location of the strongest light spot – as opposed to the dispersal signal being graded and integration simply reflecting dispersal anywhere light is focused in the membrane (and hence motility occurring on the opposite side of the cell); this seems more consistent with Figure 1. Related to this point, if you have two light sources (as in Figure 1), do the trajectories differ for bacteria close to one light source vs the other, or do they all head in the same direction, between the two sources?

*Reviewer #2:*

This paper shows that phototaxis in the cyanobacterium *Synechocystis*, a spherical cell a few μm in diameter, is just that: the migration of cells directly toward the source of light. The cells are shown to act as microlenses, generating a spot of light at the back surface of the cell, which is presumed to disperse the type-IV pili, so that the cells are pulled toward the front. The most direct and convincing experiment that shows this is the exposure of the back side of the cell to the edge of a spot of laser light, which causes a cell to move away from the light.

Some comments:

1) In the Introduction, the authors assert that a marine *Synechococcus* swims without any obvious surface appendages. Samuel and Reese showed that these cells are covered with fine filaments that extend from the inner membrane out into the external medium. (Samuel, AD, Petersen, JD, and Reese TS (2001) Envelope structure of *Synechococcus* sp. WH8113, a nonflagellated swimming cyanobacterium. BMC Microbiology 1:4.)

2) The authors refer (e.g., paragraphs two and three, subheading “*Synechocystis* cells act as microscopic spherical lenses”) to the subwavelength focusing of light. Does this mean that the focused spot is smaller in diameter than the wavelength of light, that the focused light is at a smaller wavelength than the incident light, or something else?

3) Type IV pili are known to pull rather than push cells, which might be mentioned in the Discussion. Perhaps I missed it.

4) More should be said about computation of the distribution of the optical field (FDTD), subheading “Modeling of micro-optic effects”. In particular, the reader does not know what to make of the wiggles (waves?) shown in Figure 3 and H.

5) Can we be told that the peak in Figure 3 corresponds to one of the spots evident in Figure 3? The second panel in Figure 3, with illumination from the right, shows a spot on the left and the fourth panel, with illumination from the left, shows a spot on the right. Evidently, the peak in Figure 3 corresponds to either one of those spots.

*Reviewer #3:* The main, novel conclusion of this work is that *Synechocystis* cells are lenses. Light hitting the cell on one side is focused on the other side of the cell, triggering the cell to move away from the focused spot of light and toward the environmental light source.

I was convinced by the data presented; particularly compelling were the data presented in Figure 4. Also, the experiment described in Figure 3 was very cool.

There are several improvements that could be made to make this story accessible to the greatest number of *eLife* readers. Foremost, I found the text to be very terse, with not much explanation of the biological context and little summary of the signal transduction pathway from light reception to the response by the pili. Though I endorse including the details in the Methods and legends to satisfy photoreception, phototaxis, and optical biology experts, I would encourage the authors to envision presenting the work to a much larger audience with rudimentary optics knowledge, but interest in many aspects of biology. Reviewers most often ask for manuscripts to be condensed, but in this case, I ask that the text be expanded with the goal of making the study more accessible. I have a few suggestions:

1) Figure 2 is enigmatic to me, and thus I am unconvinced by the argument that these very dark green cells do, in fact, let 80% of the light shone on them to emerge on the other side of the cell. My recommendation is to eliminate it from the main body of the paper. Is there another way to convey the data and conclusion?

2) The inclusion of the *Thermosynechococcus* data in Figure 3 was way too briefly dealt with and explained. I understand that it was included to show that the lensing was not confined to *Synechocystis*, but the introduction and presentation were a distraction rather than an enhancement. My vote would be to leave it out, perhaps mentioning it in the text.

---

## [Author Response]

*This paper reaches an interesting and novel conclusion – that phototaxis in the cyanobacterium Synechocystis operates via the cell acting as a microlenses. All reviewers felt this conclusion was well supported by the data presented, particularly the laser spot experiments in Figure 4. Nonetheless, several issues need attention before we can reach a final decision on the paper. Those identified by several reviewers are summarized below; the individual reviews follow.*

*1) The paper would benefit from a more detailed description of the signal transduction process between light absorption and the pili, as well as more details about how movement of the pili lead to cell movement. This should include a discussion of what is known about the photoreceptors themselves – e.g. what is their distribution? This could be added to the figure showing the proposed model.*

In revised paper we have focussed our Discussion only on positive phototaxis, as recommended by Reviewer 1. We must stress that even here the information available on the signal transduction pathway is very incomplete, but we can at least present a plausible model for the pathway, which we now discuss (subheading “Phototactic signal transduction in *Synechocystis*”) and present in revised Figure 5. We also explain more about pili and motility (paragraph two, Introduction).

*2) Figure 1, Figure 2 and Figure 3 all lack some details that will be important for a broad readership. Suggestions for specific things to include are in the individual comments from the reviewers.*

We have made changes to all these figures as recommended by the reviewers. The details are listed below.

*3) The Thermosynechococcus data in Figure 3 was distracting given the brevity with which it is presented. It needs either to get expanded or removed.*

We agree that the *Thermosynechococcus* data are quite preliminary and therefore we decided to remove them, together with associated discussion, as reviewers do not consider these data essential to the story. We would be interested in submitting a more complete set of data for *Thermosynechococcus* as an *eLife* Research Advance at a later stage.

*Individual reviewer comments:*Reviewer #1:

*This paper presents a very interesting idea about the mechanisms underlying phototaxis in cyanobacteria. The central idea – that the cell membrane acts as a lens to focus light on the opposite side of the cell and that this leads to changes in directed movement – is well supported by several independent analysis approaches. There are some areas in which I think the paper can be improved. Specific comments follow.*

*Light gradient experiment. The explanation of the light gradient experiment (Paragraph three, Introduction) is confusing, and I did not fully understand it until watching the movie. I would suggest including a schematic like that in the movie in the main text to make sure the light configuration is clear. It might help to state that the bacteria (presumably) try to swim towards the condenser.*

We have added a schematic similar to the one in the movie to Figure 1 (now Figure 1), and we have clarified the explanation in the text (subheading “*Synechocystis* phototaxis is based on direct light perception, rather than a biased random walk”). In fact the bacteria are “walking” in 2 dimensions on the agar surface rather than swimming, and therefore they are not able to move towards the condenser light, which is perpendicular to the surface.

*Thermosynechococcus: this is an interesting addition to the paper, but is quite underdeveloped. As a consequence it raises more questions than it answers – e.g. can they still integrate two light stimuli and if so how? Some quantification of where the spots are on the cell relative to the orientation of the light would strengthen this section. Alternatively, I think it could be left out without causing the main message to suffer.*

We agree that the *Thermosynechococcus* data are quite preliminary in nature and therefore better kept for a future publication after we have developed the work further. We have removed the *Thermosynechococcus* images and associated text.

*Dispersal mechanisms (Paragraph four, Discussion): how is Figure 4 integration?*

The cells that encounter the laser spot are presumably integrating information from the laser spot and also from the oblique light source from the bottom of the frame. We think that this is apparent from the behaviour of cells that receive just a glancing blow from the laser rather than encountering it head-on. However, since we only have a fully quantified set of data for the situation in Figure 1 we now describe the cells just as responding to multiple light sources (subheading “A model for directional light perception and phototaxis”).

*Similarly I was not sure why you assume that the dispersal signal occurs at the location of the strongest light spot – as opposed to the dispersal signal being graded and integration simply reflecting dispersal anywhere light is focused in the membrane (and hence motility occurring on the opposite side of the cell); this seems more consistent with Figure 1.*

Actually the reviewer has given a clear and succinct explanation of exactly the model that we were trying to put forward! We have clarified our explanation with help from the reviewer’s text (subheading” A model for directional light perception and phototaxis”).

*Related to this point, if you have two light sources (as in Figure 1), do the trajectories differ for bacteria close to one light source vs the other, or do they all head in the same direction, between the two sources?*

The length scales of these measurements (which are limited by the field of view of the microscope) are very short, and therefore there is no significant change in light intensity from either source across the field of view. We would therefore not expect any observable dependence of motility direction on the position within the field of view, and indeed there is no sign of such an effect (see the tracks of cell trajectories in Video 1).

Reviewer #2:

*This paper shows that phototaxis in the cyanobacterium Synechocystis, a spherical cell a few μm in diameter, is just that: the migration of cells directly toward the source of light. The cells are shown to act as microlenses, generating a spot of light at the back surface of the cell, which is presumed to disperse the type-IV pili, so that the cells are pulled toward the front. The most direct and convincing experiment that shows this is the exposure of the back side of the cell to the edge of a spot of laser light, which causes a cell to move away from the light.*

*Some comments:*

*1) In the Introduction, the authors assert that a marine Synechococcus swims without any obvious surface appendages. Samuel and Reese showed that these cells are covered with fine filaments that extend from the inner membrane out into the external medium. (Samuel, AD, Petersen, JD, and Reese TS (2001) Envelope structure of Synechococcus sp. WH8113, a nonflagellated swimming cyanobacterium. BMC Microbiology 1:4.)*

Agreed – we have modified our description of *Synechococcus* (paragraph two, Introduction). It is worth noting however that these “spicules” have only been detected in one of the motile marine *Synechococcus* strains.

*2) The authors refer (e.g., paragraphs two and three, subheading “Synechocystis cells act as microscopic spherical lenses”) to the subwavelength focusing of light. Does this mean that the focused spot is smaller in diameter than the wavelength of light, that the focused light is at a smaller wavelength than the incident light, or something else?*

We meant that the focused spot is smaller in diameter than the wavelength of light, and have clarified this point in the text (paragraph two, subheading “*Synechocystis* cells act as microscopic spherical lenses”).

*3) Type IV pili are known to pull rather than push cells, which might be mentioned in the Discussion. Perhaps I missed it.*

We now mention this point in the Introduction (paragraph two).

*4) More should be said about computation of the distribution of the optical field (FDTD), subheading “Modeling of micro-optic effects”. In particular, the reader does not know what to make of the wiggles (waves?) shown in Figure 3 and H.*

We have now extended our discussion of these simulations and the presentation of them to make them more comprehensible to non-specialists (see legend to Figure 3 and paragraph three, subheading “*Synechocystis* cells act as microscopic spherical lenses”).

*5) Can we be told that the peak in Figure 3 corresponds to one of the spots evident in Figure 3? The second panel in Figure 3, with illumination from the right, shows a spot on the left and the fourth panel, with illumination from the left, shows a spot on the right. Evidently, the peak in Figure 3 corresponds to either one of those spots.*

In fact we chose a good representative trace (close to the mean for intensity profile) and a similarly representative set of cell images – but they weren’t actually for the same cell as the profile. We have corrected this in the revised paper, which now shows in Figure 3 three views of the same cell for which the profile shown in Figure 3 was extracted. The profile was extracted from the left-hand image, proceeding around the cell perimeter in an anticlockwise direction starting from the point opposite the focused spot. This is now clarified in the revised figure legend, and we have added some additional detail to Materials and methods (subheading “Fluorescence measurements with oblique excitation”).

Reviewer #3:

*The main, novel conclusion of this work is that Synechocystis cells are lenses. Light hitting the cell on one side is focused on the other side of the cell, triggering the cell to move away from the focused spot of light and toward the environmental light source. I was convinced by the data presented; particularly compelling were the data presented in Figure 4. Also, the experiment described in Figure 3 was very cool.*

*There are several improvements that could be made to make this story accessible to the greatest number of eLife readers. Foremost, I found the text to be very terse, with not much explanation of the biological context and little summary of the signal transduction pathway from light reception to the response by the pili. Though I endorse including the details in the Methods and legends to satisfy photoreception, phototaxis, and optical biology experts, I would encourage the authors to envision presenting the work to a much larger audience with rudimentary optics knowledge, but interest in many aspects of biology. Reviewers most often ask for manuscripts to be condensed, but in this case, I ask that the text be expanded with the goal of making the study more accessible.*

Agreed – we have been through the text and tried to provide such clarification wherever possible. See subheadings “Shading is minimal in single *Synechocystis* cells” and “*Synechocystis* cells act as microscopic spherical lenses”.

*I have a few suggestions:*

*1) Figure 2 is enigmatic to me, and thus I am unconvinced by the argument that these very dark green cells do, in fact, let 80% of the light shone on them to emerge on the other side of the cell. My recommendation is to eliminate it from the main body of the paper. Is there another way to convey the data and conclusion?*

We feel that the conclusion from this figure, that a single *Synechocystis* cell absorbs only a relatively small proportion of the light passing through it, is important when considering how light intensity will vary within and around the cell, which is a theme central to our paper. Therefore we would prefer to present this figure prominently so that we can be up-front about our arguments. The reviewer does not raise any specific technical problems with the estimate of a single-cell transmission spectrum but is clearly not comfortable with the discussion. In the revised paper we have tried to explain more clearly how we derived the transmission spectrum (legend to Figure 2). The reviewer’s comment about the “very dark green cells” actually illustrates a common misconception about cyanobacteria, which perhaps reinforces the need to include the figure. Individual cells are not actually very dark green, although of course dense cultures are. Within an individual cell the local pigment concentration is quite high (as indicated in Figure 2) but the optical path-length is exceedingly short and therefore the overall light absorption is small. The direct measurement of single-cell absorption spectra is technically very challenging, and actually we are aware of only one good example in the literature for cyanobacteria. This is the pioneering work of Sugiura and Itoh (Plant & Cell Physiology, 2012) which we cite and discuss in the revised paper (lines 125-129). These authors present single-cell absorption spectra for a species of Nostoc with cell diameter a little larger than *Synechocystis*. The peak single-cell absorption is about 0.04, corresponding to a minimum transmittance of over 0.9 (ie over 90% of photons go straight through the cell). Therefore our estimate of a minimum transmittance for a *Synechocystis* cell of about 0.8 is not at all out of line.

*2) The inclusion of the Thermosynechococcus data in Figure 3 was way too briefly dealt with and explained. I understand that it was included to show that the lensing was not confined to Synechocystis, but the introduction and presentation were a distraction rather than an enhancement. My vote would be to leave it out, perhaps mentioning it in the text.*

Agreed – as discussed above we decided to leave the *Thermosynechoccus* data for a future publication. We feel that it is better not to mention it in the text if we don’t present the data.